# Spatial Transcriptomics Identifies Cellular and Molecular Characteristics of Scleroderma Skin Lesions: Pilot Study in Juvenile Scleroderma

**DOI:** 10.3390/ijms25179182

**Published:** 2024-08-23

**Authors:** Tianhao Liu, Deren Esencan, Claudia M. Salgado, Chongyue Zhao, Ying-Ju Lai, Theresa Hutchins, Anwesha Sanyal, Wei Chen, Kathryn S. Torok

**Affiliations:** 1Department of Pediatrics, University of Pittsburgh School of Medicine, UPMC Children’s Hospital of Pittsburgh, 4401 Penn Ave., Pittsburgh, PA 15224, USA; tianhao@pitt.edu (T.L.); esencand2@upmc.edu (D.E.); claudia.salgado@miami.edu (C.M.S.); chz113@pitt.edu (C.Z.); hutchinstr@upmc.edu (T.H.); sanyala@upmc.edu (A.S.); 2School of Medicine, Tsinghua University, Beijing 100084, China; 3UPMC Scleroderma Center, University of Pittsburgh, Pittsburgh, PA 15224, USA; 4UMMG Department of Pathology, Miller School of Medicine, Medical Campus, University of Miami, 1550 NW 10th Ave. #118, Miami, FL 33136, USA; 5Department of Biostatistics, University of Pittsburgh, Pittsburgh, PA 15224, USA; yil346@pitt.edu

**Keywords:** juvenile scleroderma, spatial transcriptomics, single-cell RNA sequencing, immune infiltrates, spatial domains, histopathology, morphea

## Abstract

Juvenile localized and systemic scleroderma are rare autoimmune diseases which cause significant disability and morbidity in children. The mechanisms driving juvenile scleroderma remain unclear, necessitating further cellular and molecular level studies. The Visium CytAssist spatial transcriptomics (ST) platform, which preserves the spatial location of cells and simultaneously sequences the whole transcriptome, was employed to profile the histopathological slides from skin lesions of juvenile scleroderma patients. (1) Spatial domains were identified from ST data and exhibited strong concordance with the pathologist’s annotations of anatomical structures. (2) The integration of paired ST data and single-cell RNA sequencing (scRNA-seq) from the same patients validated the comparable accuracy of the two platforms and facilitated the estimation of cell type composition in ST data. (3) The pathologist-annotated immune infiltrates, such as perivascular immune infiltrates, were clearly delineated by the ST analysis, underscoring the biological relevance of the findings. This is the first study utilizing spatial transcriptomics to investigate skin lesions in juvenile scleroderma patients. The validity of the ST data was corroborated by gene expression analyses and the pathologist’s assessments. Integration with scRNA-seq data facilitated the cell type-level analysis and validation. Analyses of immune infiltrates through combined ST data and pathological review enhances our understanding of the pathogenesis of juvenile scleroderma.

## 1. Introduction

Juvenile-onset scleroderma is a rare autoimmune disease affecting the skin, with inflammatory-driven fibrosis, leading to characteristic thickening of the skin and potential fibrosis of other organ systems [1,2]. This disease encompasses both localized scleroderma (LS) and systemic scleroderma (SSc). Clinically, systemic scleroderma typically affects internal organ systems, such as the lung [3], heart and gastrointestinal tract [4], while localized scleroderma, also known as morphea, is generally confined to the skin and underlying connective tissue, such as subcutis, fascia, tendons and muscle. Compared with the adult-onset disease, juvenile-onset localized scleroderma is more likely to result in poor outcomes, specifically regarding growth disturbances [5]. Despite clinical differences, LS and SSc share a common root, reflected in the term ‘scleroderma’ or ‘sclerotic skin’. Histologically, they are nearly indistinguishable, exhibiting similar patterns of collagen abnormalities and pockets of immune cell infiltrates [6].

Given the skin is the primary target for understanding the pathogenesis of scleroderma, many cellular and molecular studies have focused on skin tissue in adult scleroderma, including some studies based on high-throughput sequencing [7,8,9]. Although fewer studies have been conducted in pediatric scleroderma, they share similar objectives, including our studies profiling gene expression and individual cell types of skin tissue of juvenile scleroderma patients using bulk [10] and single-cell RNA sequencing (scRNA-seq) [11]. However, bulk and scRNA-seq loses the information of the spatial location of specific cells within tissues and their potential interaction. The spatial location of cells can affect the state of cells because of the different molecular signals in different spatial environments [12].

The new technology, spatial transcriptomics (10x Visium CytAssist), addresses this limitation by preserving the location of cells in histopathological slides while simultaneously sequencing the whole transcriptome [13]. Histopathological slides from skin lesion biopsies are essential for diagnosing scleroderma due to characteristic collagen abnormalities, immune cell aggregation and architectural changes commonly observed in skin lesions of scleroderma patients [14]. Spatial transcriptomics (ST) technology can measure gene expression in the original locations of the tissue slides, aiding the characterization of the disease mechanism and the identification of the molecular biomarkers correlated with histopathological features.

The 10x Visium spatial transcriptomics platform has been widely used in the cancer domain and some research exists in the cutaneous arena, namely on skin cancers, melanoma [15,16,17] and squamous cell carcinoma [18], and to a lesser extent, inflammatory/autoimmune skin conditions, such as psoriasis [19,20,21], atopic dermatitis [20,22] and infectious disease [20,23]. The skin’s heterogeneous nature, with elongated fibroblasts secreting collagen in the dermis while more spherical cells comprise the epidermis and adnexal structures, poses challenges for capturing RNA molecules across all layers. Scleroderma-affected skin differs further, with increased immune infiltration and more collagen production leading to thickening and sclerosis. These distinct histological characteristics necessitate validated procedures for preparing scleroderma skin samples for spatial transcriptomics and analyzing spatial transcriptomics (ST) data of scleroderma skin.

This study has a three-fold purpose: (1) to validate the feasibility of obtaining high-quality spatial transcriptomics (ST) data from pediatric scleroderma skin and using the ST data to distinguish anatomical structures (spatial domains), (2) to validate the comparable accuracy between scRNA and ST data and estimate the cell type composition in spatial domains by pairing ST data with scRNA-seq data from the same patient, and (3) to verify the anatomical structures in scleroderma skin identified by the ST data through pathology review and manual annotation, with a sub-aim of further characterizing immune infiltrated regions. The knowledge gained from this study will demonstrate the feasibility and accuracy of using spatial transcriptomics in pediatric scleroderma skin. The dataset will also serve as a valuable reference for future large-scale ST studies on scleroderma skin.

## 2. Results

We analyzed four pediatric scleroderma patients: one with systemic scleroderma (SSc) and three with localized scleroderma (LS). All four patients are female with age of onset ranging between 12 and 15 years old, and age at biopsy between 13 and 21 years old. All patients had active disease at the time of biopsy. The SSc patient had diffuse cutaneous disease, pulmonary hypertension, gastrointestinal dysmotility and myositis. All three LS patients had linear scleroderma of trunk/limbs, with patient c also having Raynaud’s phenomenon and patient d having deep disease with fasciitis and tendonitis. Each patient had available scRNA-seq data and paired paraffin-embedded punch biopsies for spatial transcriptomics (ST). These samples were processed and analyzed through our procedures specifically designed for scleroderma skin (Figure 1). The ST tissue processing followed the standard workflow from 10x Genomics (Pleasanton, CA, USA), with careful selection of flat tissue slides for spatial sequencing to ensure high efficiency of molecular capture. Our analysis process is composed of three key steps: (1) the detection of spatial domains: identifying distinct regions within the tissue based on gene expression patterns, (2) the integration of ST data with scRNA-seq data: comparing the accuracy of spatial transcriptomics and scRNA-seq and estimating the cell type composition in spatial domains and each spot, and (3) pathologist-guided analysis: using pathologist annotations to validate the ST data and profile the immune infiltrated regions. These steps ensure the accuracy of the data and demonstrate the potential of ST data to reveal the pathogenesis of scleroderma from skin biopsies.

### 2.1. Biological Meaningful Anatomical Structures (Spatial Domains) of Skin Can Be Identified from Spatial Gene Expression Data

The quality of the sequencing data was first evaluated with technical parameters, including unique molecular identifier counts (UMIs, total number of uniquely barcoded RNA molecules) and sequencing saturation. Then, well-known anatomical structures (including immune infiltrates) were identified from the ST data through unsupervised clustering, which demonstrates the validity of the ST data.

#### 2.1.1. Quality Control for Sequencing and Elimination of Technical Batch Effect

The four samples were labeled as a_SSc, b_LS, c_LS and d_LS for analysis (sample a with systemic scleroderma (SSc), and samples b, c and d with localized scleroderma (LS)). The spatial sequencing spots from the four slides were pooled together, and spots with UMI counts lower than 100 were filtered out. The number of spots per sample ranged from 460 to 988 and the mean number of genes per spot ranged from 4678 to 13,337. These values, along with the percentage of reads mapped to the probe set and the sequencing saturation for each sample, are summarized in Table 1. The UMI counts and the number of genes detected per spot are visualized as violin plots (Figure 2A). Notably, the sequencing saturation was approximately 95% across all four samples and the number of genes detected per spot did not significantly increase with the number of reads sequenced per spot (Appendix A). This indicates that the sequencing saturation was sufficient for detecting low-expressed genes. Only 98 genes were expressed in less than 1% of the spots. The high sequencing saturation reflects efficient UMI capture, which facilitates the detection of gene expression even in collagen-rich areas characterized by low RNA molecule counts.

It is worth mentioning that the spatial distribution of UMIs in skin tissue is not uniform. UMI counts are high in the epidermis, hair follicles and immune infiltrates, but are low in the large collagen-dense areas in the dermis (Appendix A). This is because the dermis contains wider and more elongated collagen fibers and fewer nuclei compared to other regions. Consequently, the spots in the collagen-dense areas capture less nuclei and lower UMI counts. In contrast, hair follicles, immune infiltrated regions and the epidermis have high UMI counts due to the higher cell density, allowing each spot to capture multiple nuclei.

Although the UMI counts in collagen-dense regions are low, the expression of marker genes for fibroblasts (i.e., COL1A1) can still be detected. The proportion of fibroblasts in these regions, estimated by RCTD (robust cell type decomposition) deconvolution [24] (see Section 4), is also high (Appendix A). This demonstrated that the Visium CytAssist ST platform can recover the gene expression of collagen-rich regions with a low density of nuclei and low RNA molecule counts in scleroderma skin tissue, facilitating the analysis of collagen-related tissue fibrosis using spatial transcriptomics.

The gene expression data from spots were subjected to dimensionality reduction and visualized in two dimensions with uniform manifold approximation and projection (UMAP). Initially, the gene expression data showed a distinguishable batch effect between the four slices (Figure 2B, left). Harmony, a batch correction method, was then applied to project the gene expression of cells from different batches into a shared embedding. After Harmony batch correction, the spots from the four slices were well- on the UMAP, with representation from each subject per major cluster (Figure 2B, right).

#### 2.1.2. Biologically Meaningful Anatomical Structures (Spatial Domains) Were Identified by Unsupervised Clustering Based on Gene Expression Data of Spots

Unsupervised clustering was performed on the Harmony-corrected ST data from the four samples. Initially, eight clusters were identified. One cluster, which had high mitochondrial RNA expression content and represented only 40 spots, was removed, leaving seven clusters for further analysis (Figure 2C). Marker genes for each spatial cluster (spatial domain) were identified using the Wilcoxon Rank Sum test, and the scaled expression of the top five marker genes in each cluster is shown in the heatmap (Figure 2D). Most spatial clusters had distinct marker genes corresponding to classical markers for anatomical structures (including inflammatory infiltrates), except cluster 2 and 3 (Figure 2D), which exhibited similar expression patterns. When these gene expression-based spatial clusters were mapped onto the tissue slides, cluster 2 corresponded with the epidermis, and cluster 3 corresponded to the layer right below the epidermis (Figure 2F). Consequently, cluster 3 was classified as the superficial dermis. All other spatial clusters could be classified as well-known anatomical structures (spatial domains) in skin, as shown on the tissue slides (Figure 2F).

The proportion of each cluster in each patient sample was measured (Figure 2E). We focused on cluster 0: Deep_dermis/collagen and cluster 1: Inflammatory_infiltrate. The proportion of spots belonging to cluster 0: Deep_dermis/collagen was low in samples a_SSc and c_LS, while the proportion of cluster 1: Inflammatory_infiltrate, was high in these samples. This suggests that a_SSc and c_LS have larger areas of inflammatory infiltrates and smaller areas of fibrosis compared to b_LS and d_LS, reflecting the heterogeneity in the size of immune infiltrated regions and collagen-dense regions between different patients.

Notably, the classification of the seven anatomical structures (spatial domains/clusters) was based solely on gene expression data without considering spatial relationships. This demonstrates the accuracy of the ST data and the feasibility of using data from the Visium CytAssist ST platform to classify anatomical structures (spatial domains) in histopathological slides from scleroderma skin lesions.

### 2.2. Joint Analysis of Paired scRNA Data and ST Data from Same Patients Reveals the Cell Type Composition of Spatial Domains

A significant overlap between the marker genes of scRNA clusters and ST spatial domains was observed, which demonstrated the comparable accuracy of ST data and scRNA data. The significance of the overlap of marker genes also indicated the enrichment of each type of cell in specific spatial domains. To further estimate the cell type composition of each spot in the ST data, we used RCTD (robust cell type decomposition) [24] to decompose the ST data with scRNA data as the reference. The cell type composition of each spot was also useful for the colocalization analysis in Section 2.2.2 and the pathologist-guided analysis in Section 2.3.

#### 2.2.1. Combining Spatial Transcriptomics and scRNA Data Validates the Comparable Accuracy between the Two Platforms

To compare the biological consistency between the scRNA data (Figure 3A) and ST data (Figure 3B), we assessed the marker gene overlap using the hypergeometric test (Figure 3C) [25]. The marker genes of each spatial cluster significantly overlapped with those of at least one cell type in the scRNA data (Figure 3D), indicating that ST data can detect similar marker genes as the scRNA data. This overlap also validates the accuracy of spatial domain annotation in ST data. For instance, marker genes of macrophages from scRNA-seq significantly overlap with those of ST cluster 1: Inflammatory_infiltrate (Figure 3A–C). Macrophages in the ST data also predominantly reside in cluster 1: Inflammatory_infiltrate (Figure 3E). Similarly, marker genes of T cells, B cells and endothelial cells from scRNA-seq also overlap with those in ST cluster 1: Inflammatory_infiltrate (Figure 3D) and are enriched in ST cluster 1: Inflammatory_infiltrate (Figure 3E).

In addition, the marker genes of normal skin anatomical structures in ST data significantly overlapped with those in scRNA data (Figure 3D). To be specific, the marker genes of cluster 5: Hair_follicle, cluster 2: epidermis and cluster 3: superficial_dermis significantly overlap with keratinocytes and cornified envelop cell marker genes in scRNA-seq data. The marker genes of cluster 4: Eccrine_gland significantly overlap with epithelial cell markers in scRNA data, and epithelial cells are the major component of the eccrine gland (Figure 3D). Spatial cluster 6: Arrector_pili_muscle marker genes overlap significantly with the marker genes of endothelial cells, pericytes and myelin cells in scRNA data because smooth muscle cells were not classified in scRNA data and blood vessels and neurons are considered close ‘neighbors’ to arrector pili muscle. These overlaps confirm established knowledge of cell type composition within specific anatomical structures found in skin, supporting the use of ST data for anatomical classification.

We also directly compared gene expression values between scRNA-seq and ST data. The expression of each marker gene was averaged within each cell type and spatial domain. The Pearson correlation between the averaged gene expression in ST and scRNA data is high (Appendix A). This confirms that ST data achieves similar accuracy as scRNA-seq data regarding the actual expression level of marker genes. This also allows for the deconvolution of ST data using scRNA data as the reference.

Given the resolution limitations of the Visium CytAssist platform (approximately 5–10 cells/spot), we applied RCTD (robust cell type decomposition) [24] deconvolution to decompose the ST data using scRNA-seq data as the reference. The RCTD deconvolution method extracts marker genes for each cell type from scRNA-seq data and decomposes spatial spot gene expression into cell type combinations. This method estimates the proportion of each cell type in each spot, bridging single-cell analysis with spatial transcriptomics. The cell type composition of spots corresponds with the known anatomical structures of skin (Appendix A). The averaged proportion of each cell type in each spatial domain was consistent with established knowledge (Figure 3E). Specifically, cluster 1: Inflammatory_infiltrate exhibited a high proportion of immune cells, particularly macrophages. This indicates the prevalence of macrophages in immune infiltrated regions and is consistent with the significant overlap of macrophage marker genes with those of spatial cluster 1: Inflammatory_infiltrate (Figure 3C).

#### 2.2.2. Colocalization Analysis Based on the Cell Type Composition Reveals Spatial Relationships between Immune Cells in Scleroderma Skin

We analyzed the colocalization patterns of various cell types using proportions estimated from RCTD deconvolution. A high correlation between two cell types indicates that they tend to colocalize in the same spot. Keratinocytes, melanocytes and cornified enveloped cells consistently colocalized across all four slides, which is expected as these cell types are typically located together in the epidermis (Figure 4A–D).

The colocalization between immune cells, including T cells, macrophages and B cells, was also observed (Figure 4A–D). This aligns with the known interactions between immune cells in autoimmune conditions like scleroderma [26,27]. The colocalization pattern between immune cells is weak in sample b_LS (Figure 4B), corresponding to the mild inflammation observed in the H&E image of this sample. Notably, prominent colocalization was observed between B cells, plasma cells and epithelial cells (epithelial cells constitutes the eccrine gland) in sample d_LS, suggesting strong activity of humoral immunity near the eccrine gland in this patient sample (Figure 4D). The correlation between B cells and epithelial cells in this subject (d_LS) also supports the histological findings of eccrine glands with adjacent inflammatory infiltrates (Figure 2F, d_LS).

### 2.3. Pathologist Annotation-Guided Analysis of Normal and Inflammatory Infiltrated Regions in Scleroderma Skin Lesions

The Visium CytAssist ST platform enables spatial sequencing and H&E staining on the same slide, allowing us to integrate the histopathological image and gene expression data in our analysis. A pathologist (C.M.S.), well versed in the histopathological characteristics of scleroderma skin biopsies [10], annotated the normal (including epidermis and depth divisions of the skin) and inflammatory infiltrated regions on these four slides based on the H&E-stained images, independent of gene expression (see Section 4). First, the anatomical structures identified through gene expression analysis in step 1 (Figure 2E,F) were validated by the pathologist (C.M.S.). Second, the proportion of different immune cell types within the immune infiltrated regions, as annotated by the pathologist, was profiled. The profiling demonstrated the potential of our approach to reveal an immune-related pathogenesis of scleroderma.

#### 2.3.1. Spatial Transcriptomics Data Can Identify Normal Anatomical Structures and Areas of Inflammatory Infiltrates on H&E Slides of Scleroderma Skin Tissue

To verify the validity of our data, we compared the H&E-stained images with the gene expression data per patient. The gene expression observations were highly consistent with those based on the H&E images. We used three measurements from the spatial transcriptomics (ST) gene expression data to indicate specific anatomical structures of the skin: (1) the expression of marker genes for the anatomical structures (Figure 5, first row), (2) the cell type proportions estimated by RCTD (robust cell type decomposition) (Figure 5, second row), and (3) the spatial domains classified by unsupervised clustering (Figure 5, third row). These measurements distinguished normal anatomical structures such as the eccrine gland, hair follicle, epidermis and arrector pili muscle (Figure 5A,B; Appendix A), aligning well with the corresponding H&E images (Figure 5, fourth row), validating the accuracy of ST data.

By visualizing the expression of marker genes and the proportion of different types of immune cells, we identified specific immune cell localizations in different regions of the skin. Specifically, an area near a hair follicle had a high proportion of T cells (Figure 5C), characterizing a peri-adnexal immune infiltrate enriched with T cells in sample a_SSc. In sample d_LS, an area near an eccrine gland exhibited high CD19 expression (a B cell marker) and a high proportion of B cells was observed, indicating a peri-adnexal immune region in the subcutaneous tissue enriched with B cells (Figure 5D). Meanwhile a high proportion of macrophages was observed in the collagen-rich regions of sample c_LS (Figure 5E), indicating interstitial immune regions. These different subtypes of inflammatory infiltrates are classical pathological features for scleroderma, and they are related to the disease’s pathogenesis. The cell type abundance in these inflammatory infiltrates is quantitatively measured in Section 2.3.2 with pathologist annotations.

#### 2.3.2. Combining the Pathologist-Annotated H&E Images with ST Data Reveals the Spatial Distribution of Immune Cells and Their Relationship with Other Anatomical Structures in Scleroderma Skin

To quantitatively measure the overlap between the anatomical structure observed in the H&E staining image and the spatial domains identified by gene expression, a pathologist (C.S.) annotated the anatomical structures of these four slices. The hair follicle, sebaceous gland, arrector pili muscle, eccrine gland and immune infiltrates were identified through manual annotation of spots (Figure 6A). We used the hypergeometric test to evaluate the overlap between each region annotated by the pathologist and the spatial domains identified by gene expression. All spatial domains identified by gene expression significantly overlapped with the corresponding regions annotated by the pathologist, except for clusters 0 and 3 (Figure 6B). Cluster 0 corresponds to the collagen-dense areas of the dermis, which were not specifically annotated by the pathologist as these fibers are diffusely distributed. Cluster 3 represents the superficial dermis between the epidermis and dermis, which was not annotated by the pathologist.

Although blood vessel regions were annotated by the pathologist, no specific gene expression cluster in the spatial domain analysis corresponded to blood vessels. This absence is likely because the low-resolution ST data did not capture these structures, as many small blood vessels are smaller than one spot. However, we observed significant overlap between the manually annotated blood vessel regions and cluster 1: Inflammatory infiltrate and cluster 6: Arrector_pili_muscle (Figure 6B). This suggests that blood vessels are located near these areas, with immune infiltrated regions typically demanding more blood supply. Notably, blood vessels can still be identified from ST data by viewing the expression of the PECAM1 gene, a conical endothelial cell marker, and the proportion of endothelial cells (Appendix A). The endothelial cells were also detected with cell type deconvolution and many of the blood vessels have adjacent immune cell infiltrates (Appendix A).

Our next set of analyses focused on areas of immune cell infiltration. The annotated immune infiltrated regions significantly overlapped with ST cluster 1: Inflammatory infiltrate (Figure 6C), which emphasized that immune infiltrated regions can be accurately identified based on spatial gene expression. Notably, 125 spots of manually annotated immune infiltrates were not classified as immune regions (Figure 6C). Many of these 125 spots have been found to overlap immune infiltrates within the hair follicle, eccrine gland (Figure 5C–E) and superficial dermis and thus were classified as those regions instead of immune infiltrates. The immune infiltrates in these different regions have distinct functions in the pathogenesis of scleroderma [19,28,29]. Thus, the pathologist classified the immune infiltrated regions into several subtypes based on the relationships between immune cells and other anatomical structures, including perivascular, perineural, peri-adnexal and interstitial infiltrates (Figure 6D). Each of these subtypes of immune infiltrates has distinct marker genes (Appendix A).

We measured the overlap between the spatial domains classified by gene expression and the manually annotated subtypes of immune infiltrates using the hypergeometric test. All manually annotated subtypes of immune niches significantly overlapped with cluster 1: Inflammatory_infiltrate. The peri-adnexal immune infiltrates in the dermis significantly overlapped with hair follicle regions, making them also peri-follicular regions. The peri-adnexal immune infiltrates in the subcutis significantly overlapped with eccrine glands, making them also peri-glandular regions. The perivascular immune infiltrates tended to appear in cluster 3: Superficial_dermis, the intermediate region between the epidermis and the dermis (Figure 6E).

The proportion of different immune cells in the five subtypes of immune cell infiltrated regions was profiled (Figure 6F). Interstitial immune infiltrates contained a higher proportion of macrophages compared to other immune cells (Figure 6F), indicating that many macrophages are interspersed in the collagen matrix. The proportion of different types of immune cells was also profiled in the collagen-rich regions of each sample separately (Appendix A). The proportion of macrophages was higher than other immune cells in collagen regions across all samples, except sample b_LS, which had a low abundance of immune cells overall (Appendix A). This high proportion of macrophages in interstitial collagen-rich regions indicates the potential role of macrophages in the fibrosis of scleroderma skin. This also indicates the potential colocalization and communication between macrophage and fibroblasts. The spatial distribution of three fibroblast subtypes of interest from our prior publication that were related to LS [11] (Appendix A) was visualized (Appendix A). We observed that the proinflammatory (CCL19/APOE) fibroblasts tend to locate in the inflammatory regions in sample a_SSc, c_LS and d_LS (Appendix A), and the myofibroblast-like (SFRP4/PRSS23) fibroblasts also tend to locate in inflammatory regions in sample c_LS (Appendix A). This evidence further indicates the potential colocalization between inflammatory infiltrates and scleroderma-related subtypes of fibroblasts.

We also observed that peri-adnexal immune infiltrates in the subcutis contained a high proportion of B cells and plasma cells (Figure 6F), particularly in sample d_LS. This sample exhibited two immune cell infiltrated regions in the subcutis, both with a high proportion of B cells (Figure 5D). This phenomenon indicates the high humoral immunity activity in the subcutis of patient d_LS.

This immune-centric analysis demonstrated the spatial transcriptomics platform’s ability to profile heterogeneous immune cells in different regions of scleroderma skin tissue, providing cell-level insights into immune infiltrated regions that are significant in the diagnosis and pathogenesis of scleroderma [6].

## 3. Discussion

The histopathological analysis of skin biopsies in scleroderma patients is crucial for understanding disease status and for further investigation of disease pathogenesis. Scleroderma skin tissue is unique in its properties of a mixture of fibrotic tissue with dense collagen fibers and immune cell aggregation in both pockets surrounding skin adnexal structures and dispersed throughout the collagen matrix. Spatial transcriptomics (ST) technology, which measures gene expression directly from tissue slides, offers a groundbreaking method to investigate the spatial distribution of cells and their gene expression in their native tissue context. This study is the first to leverage high-throughput ST technology to explore the spatial domains and spatial distribution of cells and their gene expression within affected pediatric scleroderma skin.

Our study demonstrates the efficacy of the Visium CytAssist ST platform for examining scleroderma skin biopsies. We identified spatial domains within the ST data that corresponded with the pathologist-annotated anatomical structures, confirming the platform’s reliability. By integrating the ST data with single-cell RNA sequencing (scRNA-seq) data, we validated the comparable biological accuracy of both platforms, further strengthening our findings.

We designed a pipeline for conducting spatial transcriptomics experiments and data analysis tailored specifically for scleroderma skin tissue (Figure 1). Our experiment achieved high sequencing saturation and successfully captured key marker genes for scleroderma skin tissue. Each step of our analysis validated the data’s accuracy, highlighting the potential of spatial transcriptomics for understanding scleroderma pathogenesis. Notably, our analysis also focused on immune infiltrated regions within the histopathological slides and found a high proportion of macrophages in collagen-rich interstitial regions, which corresponds with the previous finding on the profibrotic effect of macrophages [30,31] (Figure 6F and Appendix A). This demonstrates the Visium CytAssist ST platform’s ability to characterize these immune niches of scleroderma skin tissue at both the cellular and molecular levels.

Compared to scRNA-seq, spatial transcriptomics profiles the entire transcriptome while preserving spatial information [13], enabling the detailed mapping of immune infiltrates and their gene expression regions within their original locations in scleroderma skin slides. Immune cell infiltration of scleroderma skin is an important feature for disease diagnosis and a cornerstone feature for understanding its autoimmunity aspects. Spatial transcriptomics offers more precise information on gene expression and cell identity than manual annotation of histopathological slides. For instance, our ST data identified inflammatory infiltrates that were not detected by pathologist, such as the 238 spots within cluster 1 (Figure 6C). Upon review, the pathologist confirmed that these spots contained immune cells interspersed with collagen fibers, highlighting ST’s superior detection capabilities, especially in collagen-dense areas.

We utilized the traditional Visium ST platform (with CytAssist) with multicellular resolution, where each spot is 55 μm and contains 5–10 cells, which is currently considered ‘low resolution’ and a limitation to our study. We did observe that the correlation of gene expression between ST data and scRNA data is lower than that in our previous study comparing the two platforms for single-cell sequencing (Appendix A), which can be explained by the multicellular resolution of ST data [32]. Some computational methods have been developed to enhance the resolution of ST data [33,34,35]. In this study, we integrated scRNA with ST data to decompose ST spots and reduce the negative effects caused by the low resolution of ST data. ST platforms with subcellular resolution are commercially available now, such as 10x Genomics Visium HD, Xenium and Vizgen MERSCOPE. Although these high-resolution platforms can provide more accurate spatial information, facilitating the detection of sparsely dispersed cells [36], they suffer from low molecule-capture efficiency [37,38], which is particularly problematic in scleroderma skin tissue due to its large collagen-rich areas with low RNA molecule content. Additionally, the high cost associated with high-resolution platforms limits their applicability for large cohort studies. Therefore, we opted for the lower-resolution Visium ST platform, which provided sufficient molecule-capture efficiency for the classification of spatial domains and gene level analysis. Future advancements in high-resolution platforms may offer improved spatial information once the issues of molecular-capture efficiency and cost are addressed.

Our study only includes four samples (one SSc and three LS). We used these four samples to prove the feasibility of using spatial transcriptomics to study scleroderma skin tissue. However, we do not have enough power to detect the spatial transcriptomics differences between localized scleroderma and systemic sclerosis since we only included one SSc patient. We plan to include more samples and construct a complete spatial transcriptomics atlas of scleroderma skin tissue.

By integrating ST data with scRNA-seq data and manual tissue slide annotations, our study offers a comprehensive approach to study scleroderma skin lesions. The findings from this study underscore the potential of spatial transcriptomics in advancing our understanding of scleroderma and may pave the way for more precise diagnostic and therapeutic strategies.

## 4. Materials and Methods

### 4.1. Human Patient Skin Sample Collection

Skin biopsies were obtained from patients with pediatric-onset scleroderma, both localized and systemic, through the National Registry for Childhood Onset Scleroderma (NRCOS) at the University of Pittsburgh (PI—Torok, #PRO11060222). Two adjacent 4 mm punch biopsies were taken from affected skin (Figure 1). One specimen was preserved in CryoStor^®^ CS10 or immediately dissociated (fresh) for single-cell RNA sequencing (scRNA-seq) [11], while the other punch biopsy specimen from the same patient was formalin-fixed and paraffin-embedded (FFPE) for hematoxylin and eosin (H&E) staining for future studies including spatial transcriptomics (ST). One subject with systemic scleroderma and three subjects with localized scleroderma were included in our study. All four subjects had two adjacent biopsies for scRNA-seq and spatial transcriptomics.

### 4.2. Spatial Transcriptomics Experiment and Data Processing

#### 4.2.1. Skin Sample Processing and Spatial Transcriptomics Sequencing

The FFPE samples were used to prepare the slides for spatial transcriptomics (ST) (Figure 1). Multiple 5 μm serial sections were cut from the FFPE tissue block using the microtome. The sections were mounted on slides with an optimized floating time. The slides were then deparaffinized, stained with H&E and cover-slipped. The slides were imaged at 10x resolution using an EVOS M7000 brightfield microscope, with images stitched together to reconstruct the whole tissue area for each serial section. After imaging, one section without wrinkles was selected for Visium spatial capture, while the rejected sections were used for RNA quality checks (DV200 ≥ 30%). Since folding can affect unique molecular index (UMI) capturing during sequencing, selecting a section without wrinkles can help to solve the problem of low UMI counts in skin tissue.

Next, the coverslip was removed, and the section was assembled with the Visium CytAssist tissue slide cassette with a 6.5 mm×6.5 mm capture area. The tissue section was then destained and de-crosslinked. All four samples followed the same pipeline for tissue processing. All samples proceeded immediately to Visium probe hybridization and completion of the Visium library preparation protocol. The Human Probe Set V2 from 10x Genomics, targeting 18,085 genes, was used for hybridization. Each sample was uniquely indexed during library generation and pooled for sequencing. Sequencing was performed at the Health Science Sequencing Core at the Children’s Hospital of Pittsburgh of the UPMC Rangos Research building on an Illumina NextSeq 2000 for a target of 275M reads per sample.

#### 4.2.2. Visium Spatial Transcriptomics Data Preprocessing

The high-resolution and low-resolution H&E stained images generated with the spatial sequencing data were aligned to adjust the spatial coordinate in Loupe browser 7.0.0. The result for adjustment, together with the H&E images and raw sequencing data (fastq file), served as the input for spaceranger 2.0.1. The reference genome was GRCh38-2020-A from 10x genomics. The output of spaceranger was imported with Seurat 4.4.0 package in R for downstream analysis.

Spots lacking tissue coverage or exhibiting UMI counts below 100 were excluded from the analysis. A total of 2729 spots were retained across the four samples. The raw counts for each gene in each spot were normalized and log-transformed with the function NormalizeData in Seurat 4.4.0 with the following formula: $Normalized=\log\left(row\_counts\times 10^{4}/total\_counts+1\right)$. The top 2000 variable genes were selected using the ‘vst’ method with the function FindVariableFeatues in Seurat. The variable genes were scaled to the same mean and variance with the ScaleData function in Seurat. Principle component analysis (PCA) was performed on the highly variable genes with the function RunPCA using the default parameters in Seurat. The cells were visualized by uniform manifold approximation and projection (UMAP). Harmony correction was applied to adjust for the batch effect between the four slices with default parameters. Unsupervised clustering was performed on the corrected principal components using the FindClusters function in Seurat. The Wilcoxon Rank Sum test was performed to identify marker genes for each cluster using the FindAllMarkers function in Seurat. One cluster with a high expression level of mitochondria genes was trimmed. The remaining seven clusters, comprising a total of 2689 spots, were retained for downstream analysis. The annotation for each cluster was based on the top 20 marker genes for each cluster. The clusters were visualized on the original histological slices.

### 4.3. Single-Cell Sequencing Experiment and Data Processing

#### 4.3.1. Single-Cell RNA Sequencing Experiment

Per our prior publications [11], the cryopreserved skin samples were enzymatically digested using the Miltenyi Biotec Whole Skin Dissociation Kit (human Cat#130-101-540, Miltenyi Biotec©, Auburn, CA, USA) and dispersed using the Miltenyi gentleMACS Octo Dissociator (Miltenyi Biotec©, Auburn, CA, USA). The cell suspension was then filtered, re-suspended in PBS, mixed with reverse transcription reagents and loaded into the Chromium instrument (10x Genomics©, Pleasanton, CA, USA). The gel bead-in-emulsions (GEMs) which contained single-cell barcodes were produced. Approximately 5000–6000 cells per sample were loaded into the instrument, typically resulting in data on ~3000–5000 cells per sample. V3 single-cell chemistries were used per the manufacturer’s protocol (10x Genomics©). The resulting cDNAs were amplified, and RNA-seq libraries were generated, quantified and then sequenced (~200 million reads/sample) using the Illumina NextSeq-500 platform (San Diego, CA, USA).

#### 4.3.2. Single-Cell RNA Sequencing Data Processing

The raw fastq was processed with CellRanger to obtain gene × cell counts matrix. Quality control, normalization, selection of the variable genes, PCA, UMAP and clustering were conducted with the same steps as those applied to the spatial transcriptomics data. The step of batch correction with Harmony was skipped due to the minimal batch effect in the scRNA data. The cell types were classified with the marker genes. One cluster with high expression of mitochondria genes was trimmed. A total of 13,585 cells were retained for downstream analysis. These cells were annotated according to the marker genes for each cluster and 15 types of cells were classified at the end.

### 4.4. Joint Analysis of scRNA Data and ST Data

#### 4.4.1. Comparing Marker Genes in scRNA Data and ST Data (Multimodal Integration Analysis)

The marker genes for every cluster in the spatial transcriptomics data and single-cell data with an FDR-adjusted *p* value < 0.05 and Foldchange > 1 were used for comparison. The significance of overlap between the marker genes from each cluster in the ST data and those of each cluster in the scRNA data was measured using the hypergeometric test [25]. A significant overlap of marker genes reflects the similarity between the clusters in the scRNA data and ST data based on the gene expression level. It also reflects the enrichment of each cell type in each spatial domain. The −log10(*p* value) of the hypergeometric test was used to quantitatively measure the similarity. The expression of the overlapped marker genes was averaged in corresponding clusters in the scRNA data and ST data. The correlation between the averaged expression level of these marker genes in the ST data and scRNA data was modeled with linear regression.

#### 4.4.2. Robust Cell Type Decomposition (RCTD) Deconvolution

Robust cell type decomposition (RCTD) deconvolution was used to dissect the proportion of each cell type in every spot [24]. The paired single-cell RNA sequencing data from the same four patients were used as the reference to decompose each spatial sequencing spot into a mixture of multiple cell types. The gene expression pattern was extracted from the scRNA data by calculating the mean expression profile for each cell type. Then, the gene expression counts in each spot were fitted as a linear combination of the gene expression of multiple cell types in the scRNA data. Maximum likelihood estimation was used to estimate the proportion of each cell type in each spatial sequencing spot. The pie plot showing the proportion of each cell type in each spot was plotted with the function CARD.visualize.prop in R package CARD 1.1 [39].

### 4.5. Manual Annotation Based on H&E Image

The four slides with an H&E background were annotated by a pathologist (C.S.) familiar with dermatology pathology, specifically scleroderma skin [10], using Loupe browser 7.0.0 (10x genomics©). The spots were annotated, with special attention paid to the general architectural location (epidermis, dermis) and the immune infiltrated regions, both within the level of the dermis and surrounding structures, such as blood vessels (i.e., perivascular infiltrate), adnexal appendages, such as hair follicles and their associated smooth muscle (arrector pili), and eccrine glands [6,40]. Five immune cell infiltrate patterns were derived: perivascular, peri-adnexal/dermis, peri-adnexal/subcutaneous, interstitial and perivascular/perineural. The significance of the overlap between the spots of the annotated areas and those of the spatial cluster was measured using the hypergeometric test. The −log10(*p* value) was calculated to assess the significance of the overlap.

## 5. Conclusions

This study demonstrates the feasibility of utilizing the Visium CytAssist platform to investigate pathohistological slides from the affected skin of scleroderma patients. Firstly, the ST data can successfully distinguish spatial domains which are consistent with anatomical structures annotated by the pathologist (C.S.). Secondly, paired analysis of the scRNA data and ST data (from the same patient) reveals a significant overlap between the marker genes for each cell type identified in the scRNA-seq data and those identified in the spatial domains of the ST data. This overlap confirms the comparable accuracy between ST and scRNA-seq in measuring gene expression, supporting the rationale for mapping cells from scRNA-seq data to ST data.

Additionally, our study validated that cell types identified in scRNA-seq data can be accurately mapped to their corresponding anatomical structures in ST data with cell type deconvolution methods (RCTD), enabling the profiling of cell type composition in different spatial regions. Notably, we found that the abundance of immune cells varies across different subtypes of immune infiltrated regions, underscoring the heterogenous nature of immune infiltrates in scleroderma skin lesions.

Overall, this study highlights the potential of spatial transcriptomics as a powerful tool for understanding the cellular and molecular landscape of scleroderma skin lesions, paving the way for more precise diagnostic and therapeutic strategies.

## Figures and Tables

**Figure 1 ijms-25-09182-f001:**
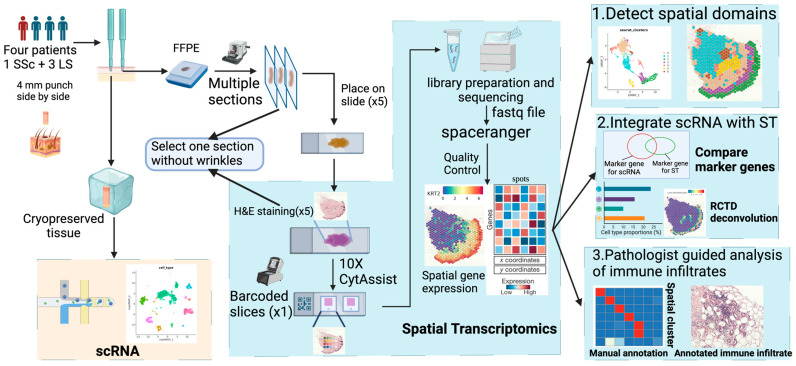
An overview of the spatial transcriptomics experiment and analysis procedures for scleroderma skin. The experiment part (**left**) includes sample collection and preprocessing. One of the adjacent biopsies was frozen and sent for single-cell sequencing. The other adjacent biopsy was preserved with FFPE for spatial transcriptomics and histologic analyses. Multiple tissue sections were H&E stained and one of them without wrinkles was selected for spatial sequencing. The H&E-stained slide was transferred to barcoded slices using the CytAssist platform. The molecular library was generated and sequenced. Raw sequencing data were processed into spot × gene matrix using spaceranger 2.0.1. The analysis part (**right**) includes three steps: (1) detect spatial domains from the spatial gene expression data, (2) integrate the paired ST and scRNA data from the same patients to validate the comparable accuracy between the two platforms, and (3) perform pathologist annotation-guided analysis of immune infiltrated regions.

**Figure 2 ijms-25-09182-f002:**
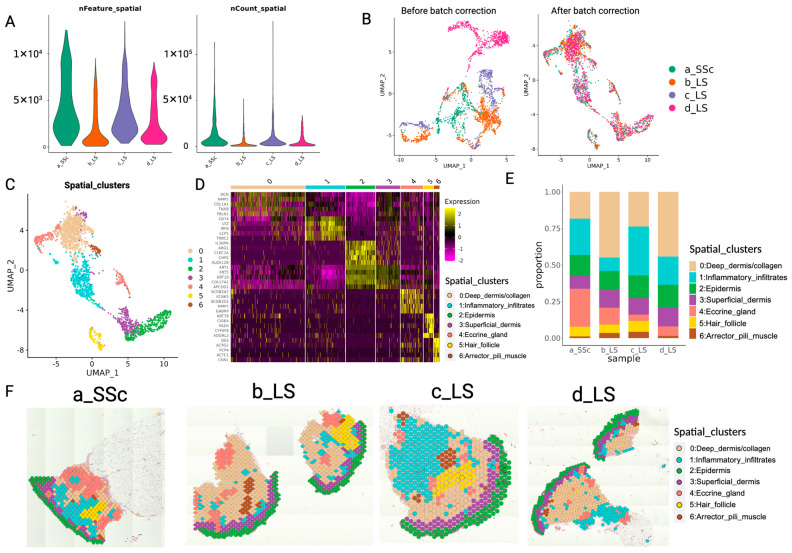
Unsupervised clustering based on gene expression identifies spatial domains (spatial clusters, anatomical structures). (**A**) Number of genes (left) and UMIs (right) in spots of four samples. (**B**) UMAP visualization of spots in four samples before batch correction (left) and after batch correction (right) (colored by sample; a_SSc, b_LS, c_LS, d_LS are four samples with systemic scleroderma (SSc) or localized scleroderma (LS)). (**C**) Unsupervised clustering of spots based on batch-corrected gene expression (colored by clusters). (**D**) Expression of top 5 marker genes for each cluster and annotation for each cluster based on gene expression. (**E**) Abundance of each spatial cluster in each sample. (**F**) Each spatial cluster was embedded on tissue slides.

**Figure 3 ijms-25-09182-f003:**
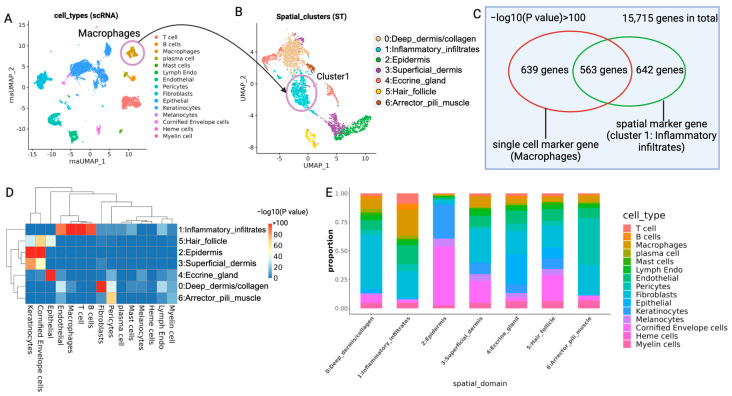
The overlap of marker genes between clusters in the scRNA-seq data and those in the ST data from paired samples. (**A**) UMAP of scRNA-seq data from four patients (cells are colored by cell type). (**B**) UMAP of ST data from four patients. Spots are colored by spatial domains. The arrow points from the macrophage cluster in scRNA data to spatial cluster 1: Inflammatory_infiltrate in ST data, indicating the enrichment of macrophages in immune infiltrated regions. (**C**) The overlap between marker genes of spatial cluster 1: Inflammatory_infiltrate and those of macrophages in scRNA data. (**D**) The overlap between marker genes of each cluster in scRNA data and ST data. The density of colors indicates the –log10(*p* value). Red indicates a lower *p* value and significant overlap. *p* values were calculated with the hypergeometric test. (**E**) The average proportion of each cell type (Y axis) in each spatial domain (X axis). The proportion in each spot was estimated with RCTD and was averaged across each cell type in all spots from the same spatial domain. The color for each cell type in (**E**) is the same as that in (**A**).

**Figure 4 ijms-25-09182-f004:**
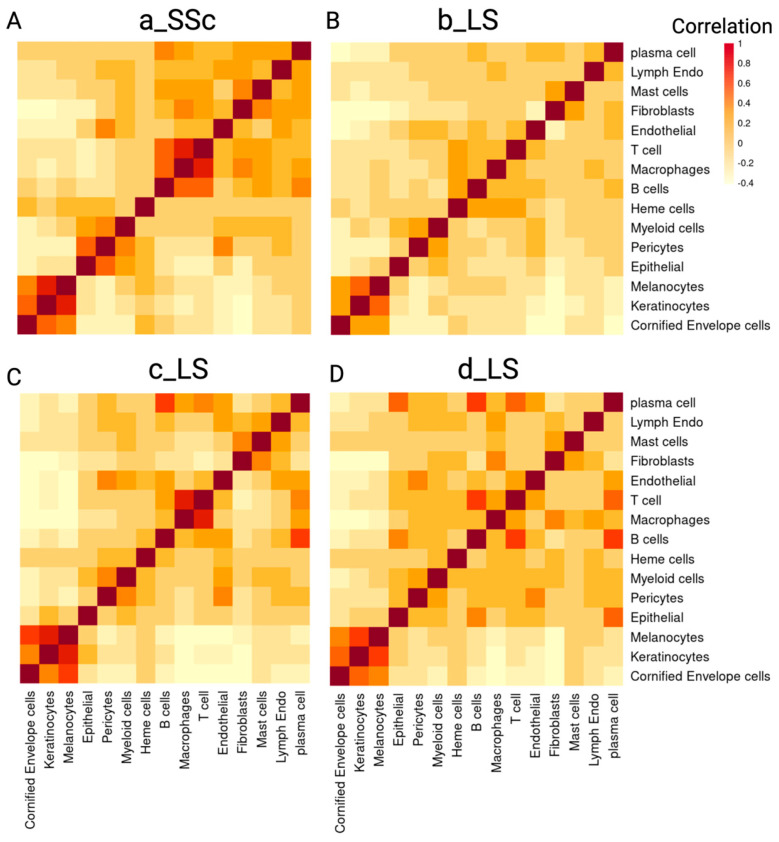
The colocalization between the different types of cells. The density of color correlates with the Pearson correlation ranging from −1 to 1. Denser orange-red color means higher positive correlation. (**A**–**D**) correspond with the four samples a_SSc, b_LS, c_LS and d_LS (SSc: systemic scleroderma, LS: localized scleroderma). The density of the squares indicates the positive spatial correlation between two types of cells corresponding to the column and row of the square. The Pearson correlation was calculated with the proportion of two types of cells in all spots.

**Figure 5 ijms-25-09182-f005:**
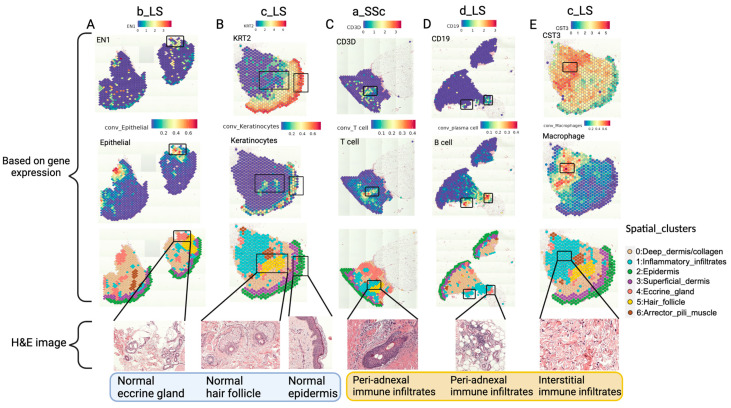
ST data distinguishes normal and inflamed anatomical structures. First row, the expression of marker genes (EN1, KRT2, CD3D, CD19, CST3). Second row, the estimated proportion of corresponding cell types (epithelial, keratinocytes, T cells, B cells, macrophages). Third row, the seven spatial clusters identified on the slides. Fourth row, the zoomed in H&E image. (**A**,**B**) The anatomical structure: eccrine gland, hair follicle and epidermis. EN1 is the marker gene for eccrine gland epithelial cells. KRT2 is the marker gene for keratinocytes which reside in the epidermis and hair follicle. Both of these annotations support the correct identification of normal structures within the skin. (**C**–**E**) Immune cells located adjacent or within the hair follicle, eccrine gland and collagen. CD3D, CD19 and CST3 are marker genes for T cells, B cells and macrophages.

**Figure 6 ijms-25-09182-f006:**
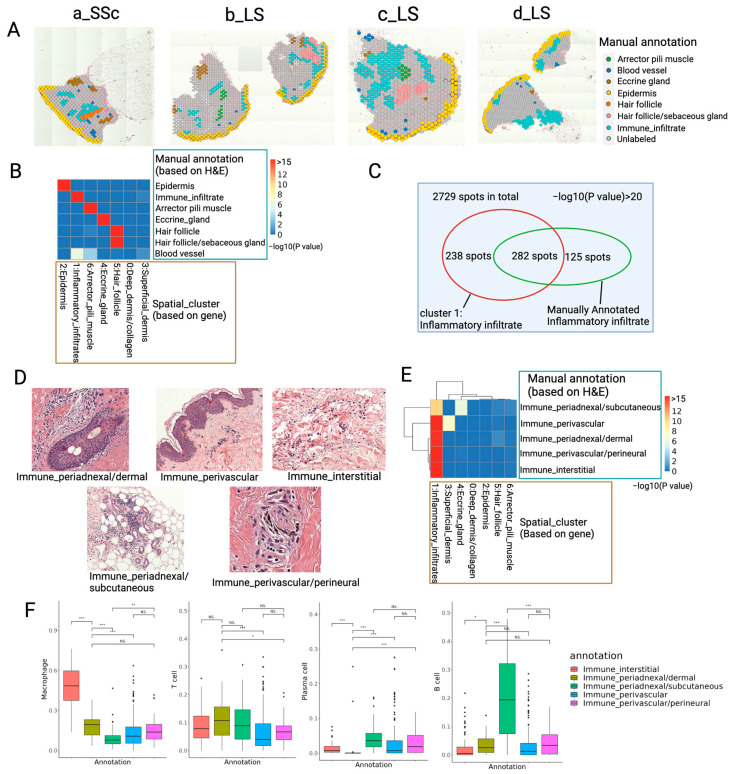
Overlap between manually annotated regions and spatial clusters (spatial domains) and profiling for subtypes of immune infiltrates. (**A**) Manually annotated regions were identified. Unlabeled regions are shown in gray. Subtypes of immune infiltrates are combined as one. (**B**) Overlap between each spatial cluster (columns) and manually annotated regions (rows). Density of heatmap indicates −log10(*p* value). Red color indicates lower *p*-value and more significant overlap. *p*-value was calculated with hypergeometric test. (**C**) Overlap of spots in manually annotated immune infiltrates and cluster 1: Inflammatory_infiltrate classified with ST data. (**D**) Examples of H&E image of 5 manually annotated subtypes of immune infiltrates. (**E**) Overlap between each manually annotated subtype of immune infiltrates (columns) and each spatial cluster (rows). (**F**) Proportion of macrophages, T cells, B cells and plasma cells in spots of different manually annotated subtypes of immune infiltrates. Y axis indicates proportion of four types of immune cells. Five boxes with different colors indicate different subtypes of immune infiltration. * indicates *p*-value < 0.05, ** indicates *p*-value < 0.01, *** indicates *p*-value < 0.001, “NS.” indicates *p*-value > 0.05.

**Table 1 ijms-25-09182-t001:** Summary of quality control metrics in four pediatric scleroderma samples.

QC Metrics	a_SSc	b_LS	c_LS	d_LS
Number of spots under tissue	460	988	564	717
Mean UMI counts per spot	13,373	4678	9716	5990
Mean number of genes per spot	4589	2032	3668	2765
Percentage of reads mapped to probe set	96.6%	96.5%	96.9%	96.6%
Saturation of sequencing	94.0%	95.0%	95.6%	95.2%

QC: quality control; a_SSc: sample a with systemic scleroderma; b_LS, c_LS and d_LS: sample b, c and d with localized scleroderma.

## Data Availability

The data from this study will be deposited on NCBI Gene Expression Omnibus as fastq files (raw) and processed data.

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
