# Peer review of "Spatial Transcriptomics Identifies Cellular and Molecular Characteristics of Scleroderma Skin Lesions: Pilot Study in Juvenile Scleroderma"

_ijms, 2024, doi:10.3390/ijms25179182_

Round 1

Reviewer 1 Report

Comments and Suggestions for Authors

This manuscript describes the first spatial transcriptomics study in juvenile scleroderma. The authors described in detail their analysis pipeline using 4 patients; 1 SSc and 3 LS patients, using the 10x Visium platform. The authors were able to identify most cell types in the skin, except for the vasculature. This limitation could be a concern, since the vascular involvement is critical for this disease. I suggest that the authors try to use the scatter pie plot that can show the cell type composition for each spot in the spatial-seq sample, to show the relative proportion of the cell types. The authors should also consider at least show the fibroblast clusters that are identified by the scRNA-seq analysis published from the same group. The reviewer is wondering whether the authors can identify where the myofibroblasts are located. Or the authors can show where the ECM are located, using the addmodulescore function. Lastly, there is large heterogeneity in the 4 skin samples. The authors should provide the information for patient characteristics.

Author Response

Please see attached response to Reviewer 1

Reviewer 2 Report

Comments and Suggestions for Authors

Comment to ijms-3129009 manuscript titled " Spatial Transcriptomics Identifies Cellular and Molecular Characteristics of Scleroderma Skin Lesions: Pilot in Juvenile

Scleroderma

This study uses spatial transcriptomics to analyze the transcriptome in one systemic scleroderma and three localized scleroderma lesions. It reveals the gene expression profiles in juvenile scleroderma.

Some specific concerns/comments are as follows:

1. Figure 5. Legend, CD19 instead of CD27 was used as marker genes for B cell.

2. Besides emphasizing the correlation of ST with scSeq, what are the transcriptomic differences between systemic scleroderma and localized scleroderma in juvenile patients?

Author Response

See attached response to Reviewer 2
